# Real-world adolescent smartphone use is associated with improvements in mood: An ecological momentary assessment study

**Matt Minich** *, Megan Moreno

Department of Pediatrics, School of Medicine and Public Health, University of Wisconsin-Madison, Madison, WI, United States of America

* mminich@wisc.edu

## Abstract

### Objective

Rates of adolescent mood disorders and adolescent smartphone use have risen in parallel, leading some to suggest that smartphone use might have detrimental effects on adolescents' moods. Alternatively, it is possible that adolescents turn to smartphone use when experiencing negative mood. The purpose of this study was to explore the relationship between adolescent smartphone use and mood using a longitudinal methodology that measured both in real-time.

### Method

This study used an Ecological Momentary Assessment (EMA) procedure completed by 253 12-17-year old participants from across the United States. Participants received short surveys delivered to their smartphones at random points throughout the day. Measures included real-time, in-situ assessments of smartphone use, current mood, and mood before smartphone use.

### Results

Based on tests of a multilevel regression model, adolescent moods were positively associated with smartphone use ($\beta = 0.261$, $F(1,259.49) = 19.120$, $p < 0.001$), and that mood was positively associated with the length of phone use sessions (*length of phone use* $\beta = 0.100$, $F(1, 112.88) = 5.616$, $p = 0.020$). Participants also reported significant changes in mood during phone use, such that moods before phone use were significantly lower than moods during phone use ($M_{Change} = 0.539$, $t(2491) = 23.174$, $p < 0.001$). Change in mood (mood before minus mood during phone use) was positively associated with the length of smartphone use sessions ($\beta = 0.097$, $F(1,122.20) = 4.178$, $p = 0.043$), such that participants who had a higher change in mood were more likely to report a longer length of smartphone use.

**Data Availability Statement:** The data used for this analysis is available for download to qualified researchers through the Technology & Adolescent Mental Wellness program (TAM) data consortium at https://tamprogram.org/tam-data-consortium/.

The dataset is officially titled "Youth Health and Social Media 2019 – Objective 3 dataset"

**Funding:** This study used data from a larger project that was funded by a stand-alone research agreement between Facebook's Youth Research Fund (2018-2020, Facebook, Inc.) and PI MA Moreno. The funder had no role in study design, data collection and analysis, decision to publish, or preparation of the manuscript.

**Competing interests:** The authors have declared that no competing interests exist.

## Conclusions

Findings suggest that adolescent smartphone use is positively associated with mood. This finding may suggest that adolescents use smartphones for mood modification, which aligns with an understanding of smartphone use as potentially addictive behavior.

## Introduction

In the 15 years since the iPhone and Android OS were introduced to the market, smartphone use has become nearly ubiquitous among U.S. adolescents—about 95% now report having access to a personal smartphone and 46% say they are connected to the internet "almost constantly" [1]. Rates of adolescent depression [2,3], self-harm [4], and suicide [5] have also risen over this period, leading some to suggest that smartphone use is detrimental to adolescents' emotional well-being [6]. This fear is widely shared among parents [7], and some scholars have called for policies to regulate or restrict adolescent use of smartphones and social media [8,9].

There is evidence that some specific patterns of digital technology use are associated with adolescent mood problems. For example, adolescents who exhibit problematic or compulsive smartphone use are more likely to be depressed [10], experience more mood disturbances [11], and have less-developed emotion regulation skills [12]. Recent reviews have also associated symptoms of depression with adolescent social media use [10,13], but authors of these studies noted that effects vary greatly across studies and might be related to some patterns of social media use but not others. Indeed, studies that distinguish between passive (i.e. scrolling through social media feeds without posting or interacting) and active patterns of social media use generally find that depressed moods are more strongly associated with passive use [14]. Thus, there is evidence that the moods and emotional health of adolescents are negatively associated with *specific types* of smartphone use: problematic use and passive social media use.

Research into the broad effects of smartphone use (i.e. effects that are not constrained to a specific application or pattern of use) has so far provided mixed and often contradictory conclusions. Some scholars have directly associated the recent rise in mood problems with increases in technology use [6,9,15], but others report that technology use has positive [16] or neutral [17] effects. Some recent reviews and meta-analyses have observed small negative effects of smartphone use on adolescent mental health [18,19] while others suggest effects are mixed and uncertain [20,21]. Adolescents themselves are also divided on the issue—24% of respondents to a recent poll said digital media use had "mostly negative" effects on their lives while 31% reported "mostly positive" effects [22]. Thus, it remains unclear whether smartphone use in general has any effects on adolescents' moods or mental health. This may be due to methodological limitations in the existing literature [20], three of which are reviewed below.

### Overreliance on recalled estimates of smartphone use

Research into the effects of technology use of all kinds—from smartphone use to social media use to overall screen time—often measures quantity or frequency of use through one-time, self-reported recalled estimates. For example, past studies have asked participants to estimate the total number of minutes they spend on their phone each day [23] or week [24], or to estimate how many times they pick up their phones during the day [25]. A recent systematic review and meta-analysis found that measures like these have limited validity because they are only moderately correlated with more objective measures of screen time use [26]. Further, people with

lower psychological well-being have been found to overestimate their smartphone and social media use when asked to offer recalled estimates, falsely inflating the associations between smartphone use and mental health concerns [24]. These problems may be exacerbated among adolescents, who overestimate the amount of time they spend on social media [27].

Recalled estimates are often used to measure smartphone use in surveys, which are the most common data collection techniques used in this area of research [20,28]. Meta-analyses and systematic reviews that have found associations between technology use and mood [10,13] have drawn primarily from studies that used recalled estimates, so it is possible that their findings were affected by participants' recall bias. Recognizing this as a serious gap in the literature, scholars have recently called for the use of measures that are less sensitive to the biases associated to estimation and recall [29]. One possible approach is the use of *in-situ* measures of smartphone use like those facilitated by Ecological Momentary Assessment (EMA) methods. These measures capture smartphone use in the moments it occurs, which facilitates easier estimations and greatly reduces the recall bias [30]. Indeed, research has found that *in-situ* measures of adolescent social media use delivered much smaller estimates of smartphone use than self-reported recalled estimates [31]. Thus, the current understanding of the interaction between smartphone use and adolescent mood is largely built around measures of smartphone use that are known to be biased.

## Insensitivity to within-participant associations

Another reason that scholars struggle to find consistent associations between adolescent smartphone use and mood may be that these associations exist at a within-participants level on a timescale that cross-sectional data cannot capture. A recent review of research into the effects of digital technology use on adolescent well-being suggested that technology use is more likely to have transient short-term effects on adolescents' emotions than on long-term emotional health outcomes [20]. Such effects would be mostly invisible to studies of cross-sectional data, which are fixed at a single moment in time. Scholars have thus called for more experimental and longitudinal research [19,20,29].

The limited experimental and longitudinal research performed so far suggests that digital technology use can indeed have short-term effects on emotional states, but the strength and valence of observed effects varies depending on the type of technology use. For example, experiences of acceptance on social media (such as receiving a "like") tend to increase activity in brain regions associated with reward [32,33] while rejection experiences interact with networks associated with depression and negative affect [34,35]. Further, experimenters who induced passive social media use found it immediately increased boredom [36]. In longitudinal studies, passive social media use has been found to precede and predict depressed mood among adolescents [37], while intentional management of a Facebook profile was found to predict improvements in self-esteem [38]. Notably, these studies manipulated the use of specific smartphone apps or styles of smartphone use—little is known about the within-participant effects of smartphone use in general.

## Lack of evidence for temporal precedence

Cross-sectional designs are also limited because they do not allow researchers to determine whether smartphone use precedes or follows its supposed effects [29]. Temporal precedence is considered essential to causation—if it cannot be established, any observed relationships must instead be considered correlational. If smartphone use is found to have short-term, within-participant associations with mood, this might be explained in two ways. Smartphone use could cause changes in mood (e.g. using a smartphone causes adolescents to feel depressed), or it

could be that moods drive adolescents to use their smartphones (e.g. adolescents are more likely to use their phones when they feel depressed).

## The present study

In sum, inconsistent findings about the relationship between smartphone use and adolescent mood may be attributable to methodological limitations. Much research in this area has relied on data collected through cross-sectional surveys, which often use biased measures, are insensitive to within-participant effects, and cannot be used to establish causality. Experimental data in this field is also limited because it has generally focused on specific patterns of smartphone use and because it is usually conducted in a controlled environment. These approaches are unable to detect the transient, moment-to-moment effects of real-world smartphone use on mood, which may be more reliable than associations with longer-term outcomes like life satisfaction and well-being [20]. Thus, the current literature is not able to verify or refute claims that smartphone use might be harmful to adolescents' mental or emotional health.

The present study sought to address these gaps by using an Ecological Momentary Assessment (EMA) data collection protocol, which allowed for *in-situ* measures of both adolescent smartphone use and mood. This approach allowed the study team to capture the interactions between these variables in real time in a real-world setting and afforded the ability to examine within-participant associations while controlling for differences between participants. Further, this protocol allowed researchers to compare adolescent mood during smartphone use with moods before smartphone use, allowing for inference about temporal orientation.

**Study objective.** The goal of this study was to test whether real-world smartphone use might have reliable but transient effects on adolescents' moods, using an *in-situ* Ecological Momentary Assessment approach.

*RQ1*: *Does adolescent smartphone use have short-term, within-participant associations with mood?*

*RQ2*: *Do adolescents report changes in mood after starting smartphone use?*

## Materials and methods

The study was conducted in five independent waves between October of 2019 and March of 2020 using an ecological momentary assessment (EMA) approach that delivered assessments directly to participants' smartphones. This approach was chosen for three reasons: First, EMA procedures measure participants' experiences in real time, which minimizes recall bias [39]. This makes EMA well-suited to measure experiences that shift over time like smartphone use and mood, and to capture any short-term, within participant associations between smartphone use and mood. Second, the longitudinal nature of EMA data allows researchers to make inferences about temporal precedence, which is needed to fill some gaps in the literature [29]. Finally, the nested nature of EMA data accommodates multilevel modeling procedures, which allow researchers to discern stable between-participant associations from transitory within-participant associations. A recent systematic literature review identified only 13 studies that had used EMA approaches to assess technology use and mood [40], but these studies captured their measures using phone calls, computers, or diaries. None used mobile technologies, and all measured experiences at regular time periods, which allows for recall bias. To minimize this bias, authors of the review suggested future work conduct EMA over brief periods of time. This study leveraged the recommendations of that review and advances the field by delivering assessments directly to participants' smartphones at random increments.

## Procedure

Participants were recruited for participation using Qualtrics and Facebook advertising. To qualify for the study, participants needed to be between the ages of 12 and 17, be able to speak English, and have access to a smartphone that could send and receive text messages that contained pictures. After providing assent and parental consent, participants completed a brief online survey and were invited to enroll in the EMA portion of the study. Participants who completed the initial survey received $10 and participants who completed 75% of assessments during the EMA period received an additional $40. These recruitment procedures and the protocol were approved by the relevant university IRB.

The EMA procedure consisted of 30 assessments (short surveys) delivered via text message over a six-day period. Assessments were delivered at random intervals during waking hours (9 a.m. to 9 p.m.), with a minimum of one hour between texts. Responses that were not submitted within an hour were excluded from analysis. EMA assessments were designed to facilitate response through text messages. Instructions for completing the assessments were delivered to each participant via text message before the first assessment was sent.

**Ethics statement.** This study was approved by the Social/Behavioral Sciences Institutional Review Board at the University of Wisconsin-Madison (approval #2018–1591). Written parental consent and participant assent were obtained through an online portal prior to the data collection procedure.

**Participants.** A total of 253 participants responded to 92.77% of the 6,780 assessments delivered during the study period. Of the 253 participants originally enrolled in the study, 52 were dropped from the study before analysis because they responded to fewer than 23 of the 30 assessments delivered during the study period, removing 827 assessments from analysis. An additional 32 observations were removed because participants used the sample text provided with EMA instructions, including all observations from one participant who only provided sample text. Eight participants were removed for failing to complete all relevant survey measures, and one participant was removed for reporting impossible durations of phone use, indicating poor data quality. The final sample included $N$ = 191 participants and 5,192 observations. The mean age of participants was 14.98 (SD = 1.41), and the sample was predominantly white (82.16%), non-Hispanic (90.18%), and cis-gendered female (70.18%).

**Measures.** This study sought to determine whether smartphone use had any short-term associations with adolescent moods and whether adolescent moods changed during smartphone use. Because both smartphone use and mood can be affected by other factors, we also considered the timing of individual assessments and measured individual differences. Thus, our measures can be sorted into four categories: assessment timing, individual differences, smartphone use, and mood.

**Assessment timing.** Past research suggests that adolescent moods exhibit certain predictable patterns over time, with adolescents reporting mood improvement over the course of the day [41] and more positive affect on weekend days [42]. Thus, we used the timestamp associated with each assessment to denote the *time of day* and *day of week* in which the assessment was completed. Because past work notes an important distinction between weekdays and weekends, we categorized *day of week* into weekday (Monday–Friday) and weekend (Saturday–Sunday) days.

**Individual differences.** In addition to demographics, we included two individual differences as level 2 (participant-level) variables in the model that are known to exert effects on adolescent smartphone use or mood. All these measures were assessed in the short survey given before the EMA portion of the study. Because recent research suggests socioeconomic status are related to negative effects of smartphone use [41], we assessed whether participants

received *free or reduced lunch* at their schools. We also assessed symptoms of *depression* because this mood disorder is increasingly common among adolescents [2] and is characterized by its negative effects on mood. *Depression* was assessed using the PHQ-9 depression scale (Kroenke & Spitzer, 2001), which is a validated self-report assessment for symptoms of depression. An example item from the scale is "How often have you been bothered by the following over the past two weeks: Feeling down, depressed, or hopeless?" Participants rated the items on a scale from 0 = "Not at all" to 3 = Nearly every day. This scale achieved an acceptable level of inter-item reliability $\alpha$ = .879.

**Smartphone use.** We captured two measures of smartphone use: *using smartphone* and *length of phone use*, as level 1 (assessment-level) variables.

*Using smartphone* captured whether a participant was on their phone at the moment they received the EMA assessment via text message. It was measured with a single item in the EMA prompt: "When you got this text, were u already on ur phone?" A "yes" response to this item was coded as 1, and a "no" response was coded as 0.

*Length of phone use* was measured only when participants reported they were on their phones and it captured the length of that specific phone use session. This was measured with a follow-up to the *using smartphone* prompt: "If yes[on phone], for how long (min)?"

**Mood.** Assessments captured two measures of adolescent mood: *current mood* and *mood before smartphone use*. From these, we also calculated a third mood measure: *change in mood*.

*Current mood* describes the mood of participants at the moment they responded to each EMA assessment. It was assessed with a single item: "Rate ur mood right now (1–7)". Ratings of 1 indicated the lowest possible mood and ratings of 7 indicated the highest.

*Mood before smartphone use* was only captured when participants said they were using their smartphones at the moment they received an EMA assessment. In this case, they were asked to respond to the prompt "Rate ur mood before using ur phone (1–7)".

## Analysis

Our analyses tested two questions. First, we tested whether adolescent smartphone use was associated with moods in the short-term at the within-participant level. Second, we evaluated whether adolescents tended to say that their moods had gotten better or worse after using their phones. Analyses were conducted in R version 4.3.2 (2023-10-31) [42] using the lme4 package [43].

### RQ1: Does smartphone use have short-term associations with adolescent mood?

Our first model tested whether adolescents' smartphone use had short-term associations with moods while controlling for the effects of individual differences. We fit a linear mixed-effects model (LMEM) with *current mood* ratings as the outcome and *age*, *race*, *gender*, *free lunch*, *depression*, *time of day*, *weekday/weekend*, *using smartphone*, and *length of smartphone use* as predictors. The model included by-participant random intercepts to account for intra-participant correlations in mood ratings, and by-participant random slopes for both *using smartphone* and *length of smartphone use* to account for participant level differences in the strength of those effects.

### RQ2: Do adolescents report changes in mood after starting smartphone use?

To test whether participants' moods changed during phone use, we made a subset of our sample that only included those EMA responses for which adolescents said they were using their phones. We also adjusted the structure of the data so that both *current mood* and *mood before smartphone use* were denoted as *mood*, with a separate *time* variable noting whether the mood was reported before or during smartphone use. We tested the fit of a linear mixed effects

model in which *mood* served as the outcome variable and *time of day*, *weekday/weekend*, *age*, *gender*, *race*, *ethnicity*, *free lunch*, *depression*, *time*, and *length of smartphone use* served as predictors. This model included a by-participant random intercept to account for intra-participant correlations in mood, and by-participant random slopes for the effects of *time* and *length of smartphone use*.

## Results

Our analyses produced two major findings. First, we found that adolescents reported better moods when they were using their phones. Second, responses to EMAs received during phone use indicated that adolescents' moods improved during smartphone use.

### Individual differences, smartphone use and mood scores

Among the participants, 33.26% reported that they received *free or reduced lunch* at their school. Adolescents reported a mean *depression* score of 1.66 ($SD$ = 0.59), which is well below recommended diagnostic cutoffs for depression, which range from 8 to 11 [44]. Participants indicated that they were on their phones in 46.76% of assessments and reported an average *length of phone use* of 11.93 minutes ($SD$ = 22.48). Participants' average *current mood* rating was 5.31 ($SD$ = 1.33), and the average *mood before smartphone use* was 4.93 (SD = 1.46).

### Finding 1: Adolescent smartphone use had a positive short-term, within-participant association with mood

Adolescents reported higher mood ratings in response to EMAs they received while using their phones (*using smartphone $\beta$* = 0.332, *F(1,182.1)* = 35.374, *p* < 0.001). Differences in mood ratings before phone use and during phone use are illustrated in Fig 1 below. Mood ratings were also negatively associated with *depression*, such that participants with stronger

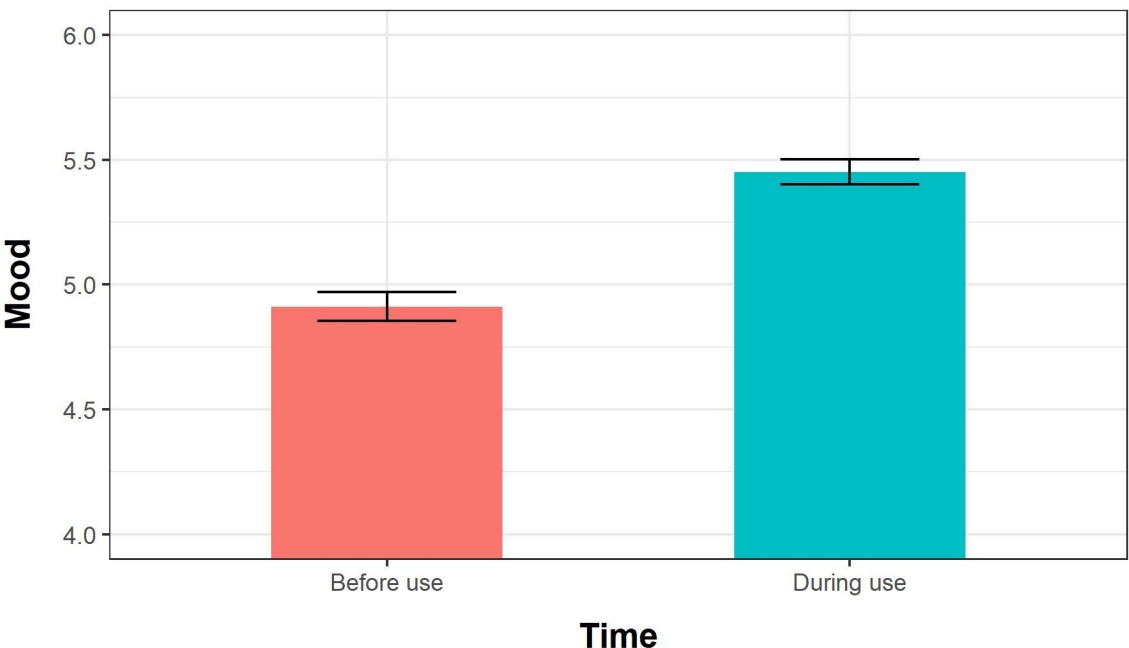

**Fig 1. Moods reported before and during smartphone use.** Note: Error bars represent 95% confidence intervals estimated around means calculated from raw data.

depression symptoms reported lower moods throughout the study ($\beta$ = -.253, $F(1,179.2)$ = 20.210, $p < 0.001$).Participant moods were positively associated with *time of day*, indicating that participants tended to report higher moods at later hours ($\beta$ = 0.039, $F(1,5043.6)$ = 50.136, $p < 0.001$). No other variables showed significant associations with *current mood*. Complete results, including variances of random effects, are described in Table 1 below.

### Finding 2: Adolescent moods improve during smartphone use

Among those assessments taken when participants were using their phones, adolescents reported higher moods during phone use than before phone use (*time $\beta$* = 0.513, $F(1,179.9)$ = 116.137, $p < 0.001$). Again, participant *depression* was negatively associated with mood ratings ($\beta$ = -0.374, $F(1,178.5)$ = 35.004, $p < 0.001$), and *time of day* was positively associated ($\beta$ = 0.024, $F(1,4568.3)$ = 19.484, $p < 0.001$). Complete results, including variances of random effects, are described in Table 2 below.

## Discussion

The goal of this study was to test whether real-world smartphone use might have reliable but transient effects on adolescents' moods, using an *in-situ* Ecological Momentary Assessment approach. Toward that end, we explored two research questions. First, we investigated whether adolescent smartphone use had short-term, within-participant associations with mood. Second, we tested whether adolescents reported changes in mood after starting smartphone use. Results showed that adolescents reported better moods when they were using smartphones

**Table 1. Effects of participant-level and assessment-level effects on current mood ratings.**

| FIXED EFFECTS | $\beta$ | df | df_resid | F | p |
|---|---|---|---|---|---|
| TIME OF DAY | 0.039*** | 1 | 5,043.6 | 50.136 | < 0.001 |
| WEEKEND | -0.005 | 1 | 4,937.1 | 0.024 | 0.878 |
| AGE | | | | | |
| 14 | 0.051 | 1 | 176.6 | 0.091 | 0.763 |
| 15 | -0.176 | 1 | 175.0 | 0.981 | 0.324 |
| 16 | 0.059 | 1 | 177.6 | 0.122 | 0.727 |
| 17 | -0.148 | 1 | 177.2 | 0.659 | 0.418 |
| RACE | | | | | |
| BLACK | 0.228 | 1 | 178.1 | 1.530 | 0.218 |
| ASIAN | 0.072 | 1 | 170.6 | 0.028 | 0.867 |
| NATIVE | -0.225 | 1 | 181.2 | 0.369 | 0.544 |
| OTHER | -0.780 | 1 | 164.3 | 1.716 | 0.192 |
| HISPANIC | 0.172 | 1 | 173.4 | 0.861 | 0.355 |
| GENDER | | | | | |
| MALE (CIS) | 0.176 | 1 | 180.9 | 2.053 | 0.154 |
| OTHER | 0.086 | 1 | 176.7 | 0.020 | 0.192 |
| FREE LUNCH | 0.062 | 1 | 177.0 | 0.276 | 0.600 |
| DEPRESSION | -0.253*** | 1 | 179.2 | 20.210 | < 0.001 |
| ON PHONE | 0.332*** | 1 | 182.1 | 35.374 | < 0.001 |
| | | | | | |
| RANDOM EFFECTS | $\sigma^2$ | df | | | |
| INTERCEPT | 0.608 | 190 | | | |
| ON PHONE | 0.372 | 190 | | | |

\* = p < 0.05

\*\* = p < 0.01

\*\*\* = p < 0.001.

**Table 2. Effects of participant-level and assessment-level effects on changes in mood (current mood ratings–before use mood ratings).**

| FIXED EFFECTS | β | df | df_resid | F | p |
|---|---|---|---|---|---|
| TIME OF DAY | 0.024*** | 1 | 4568.3 | 19.484 | < 0.001 |
| WEEKEND | 0.011 | 1 | 4500.0 | 0.126 | 0.723 |
| AGE | | | | | |
| 14 | 0.099 | 1 | 174.1 | 0.276 | 0.600 |
| 15 | -0.137 | 1 | 173.2 | 0.480 | 0.489 |
| 16 | 0.085 | 1 | 176.1 | 0.204 | 0.652 |
| 17 | -0.096 | 1 | 173.1 | 0.224 | 0.626 |
| RACE | | | | | |
| BLACK | 0.236 | 1 | 170.9 | 1.242 | 0.267 |
| ASIAN | 0.192 | 1 | 165.2 | 0.163 | 0.687 |
| NATIVE | -0.340 | 1 | 188.2 | 0.654 | 0.420 |
| OTHER | -0.876 | 1 | 150.8 | 1.858 | 0.175 |
| HISPANIC | 0.061 | 1 | 166.2 | 0.088 | 0.767 |
| GENDER | | | | | |
| MALE (CIS) | 0.140 | 1 | 179.1 | 1.007 | 0.317 |
| OTHER | 0.247 | 1 | 166.2 | 0.139 | 0.710 |
| FREE LUNCH | 0.085 | 1 | 172.2 | 0.421 | 0.517 |
| DEPRESSION | -0.374*** | 1 | 178.5 | 35.004 | < 0.001 |
| TIME | 0.513*** | 1 | 179.9 | 116.137 | < 0.001 |
| PHONE MINUTES | 0.040 | 1 | 161.7 | 0.835 | 0.362 |
| | | | | | |
| RANDOM EFFECTS | $\sigma^2$ | df | | | |
| INTERCEPT | 0.655 | 187 | | | |
| TIME | 0.261 | 187 | | | |
| PHONE MINUTES | 0.931 | 187 | | | |

\* = p < 0.05

\*\* = p < 0.01

\*\*\* = p < 0.001.

than when they were not and reported that their moods improved during smartphone use.. These results contribute to the ongoing conversation about adolescent smartphone use in two ways. First, our study addresses the methodological limitations of past work to offer new insight into the transient, short-term relationship between smartphone use and adolescent mood. Second, findings align with past research to suggest that adolescents may sometimes use smartphones to manage or modify their moods.

## Implication 1: Smartphone use has positive within-participant association with mood

The results of this study add much-needed nuance to the conversation about smartphone use and adolescent mental health. Extant research in this area has been limited by methodological factors, such as reliance on recalled estimates to measure smartphone use and the dominance of cross-sectional designs. At worst, this literature is biased by overestimated and underestimated measures of smartphone use [26]. At best, it can only provide insights into the relationship at the between-participant level (e.g. whether across adolescents, those who use smartphones are more or less depressed). Some scholars have proposed that smartphone use has more pronounced effects on short-term affect and very little work has explored the ways adolescents' smartphone use interacts with their moods in the real world. This study fills that

gap to find that the relationship between adolescent mood and real-world smartphone use is a positive one.

This study's design addressed the methodological limitations of past work in three ways. First, the EMA protocol assessed smartphone use in real-time, using the smartphone itself to deliver an *in-situ* measurement. This measure did not depend on participants' recalled estimations, which are known to be over or underestimated. Second, the longitudinal, repeated-measures nature of this design made it possible to observe within-participant associations that cannot be captured with cross-sectional, between-subject designs. Third, assessing both *current mood* and *mood before smartphone use* made it possible to establish that smartphone use preceded changes in adolescent mood, which strengthens the argument for causal effects.

Importantly, the multilevel modeling procedure used in this analysis allowed us to estimate within-participant associations while accounting for any between-participant associations. The possibility that adolescents who tend to have better moods check their phones more often and that any variance in mood attributable to differences in individual phone use rates was accounted for by our random effects structure. Further, adolescents' moods improved during smartphone use. Though we do not claim that these results indicate smartphone use had a causal effect on mood, these two findings together suggest that the positive association between smartphone use and adolescent mood emerges during smartphone use.

## Implication 2: Adolescent smartphone use may be motivated by mood modification

The observation that adolescents experience better moods and mood improvements during smartphone use could provide some insights into adolescents' motivations for smartphone use. Mood Management Theory [45] proposes that people select media experiences that help them prolong their positive moods and alleviate their negative moods. This theory has proven effective at explaining a variety of media behaviors [46], including the entertainment choices of adolescents [47], but it has not often been applied to smartphone use. Still, there is some evidence that smartphone affordances can serve a mood management function. For example, text messaging has been shown to alleviate the negative mood brought on by social rejection in adolescents [48] and adolescents reported in surveys that they used social media to manage negative moods during the COVID-19 pandemic [49].

Use of a behavior for mood modification is an important component of behavioral addiction, along with the development of tolerance and the experience of withdrawal [50]. Addicts seeking mood modification engage in behaviors "as a way of producing a reliable and consistent shift in their mood state" (50 p. 194) to cope with negative experiences. Surveys have found that adolescents report more mood modification from smartphone use than adults, and adolescents are also more likely to exhibit problematic smartphone behaviors [51]. Further, adolescents who exhibit problematic smartphone use have been shown to have less functional mood regulation skills, suggesting they may rely on smartphone use as a coping mechanism [12].

Thus, the positive associations between smartphone use and adolescent moods are not necessarily incompatible with the contention that heavy smartphone use can be harmful to adolescent mental health. These results do preclude the possibility of a linear relationship, however—because individual smartphone use sessions tend to coincide with mood improvements, it is unlikely that the mood problems associated with problematic smartphone use are caused by an accumulation of negative smartphone experiences. Instead, these results align with an understanding of smartphone use as a mood management tool that may sometimes be employed in a dysfunctional manner.

Recognizing that this interpretation of smartphone use might be seen as pathologizing a common behavior [52], we caution that adolescents who use smartphones to modify their moods are not necessarily engaged in problematic or addictive use. Both "addicted" and non-addicted smartphone users have reported turning to their smartphones for temporary mood boosts [53], suggesting that this may sometimes be a healthy or at least harmless coping mechanism. In addition to use for mood modification, the components model of behavioral addiction suggests that addictive behaviors have five additional components: increased salience of the behavior, development of tolerance to the mood-modifying effect, withdrawal symptoms in the absence of the behavior, conflict with others as a result of the behavior, and relapse to problematic patterns of the behavior after attempts at change [50]. Though the findings of this study suggest that adolescents can use smartphones to improve their mood, they do not suggest that doing so necessarily generates addictive patterns of behavior or displaces healthier coping mechanisms. In fact, it is possible that smartphone use can sometimes serve as a healthy way for adolescents to manage difficult moods.

## Limitations and recommendations for future research

This study offered much-needed insights into the real-world adolescent experience of smartphone use, but its findings are not without limitations. Though a major strength of our EMA approach was that this method minimizes participant recall [39], some measures did technically still rely on recalled estimations. Though *length of phone use* and *mood before smartphone use* were both assessed in or near the moment of smartphone use, both required participants to estimate an experience that had already occurred. Thus, both are conceivably vulnerable to the biases associated with estimation and recall. Importantly, these reports may also be vulnerable to other biases associated with self-report [54]. For example, some participants may have underreported *length of phone use* due to social desirability pressures.

Second, because the purpose of this study was to assess the effects of smartphone use in general, our analyses did not consider the types of smartphone use. Smartphones are versatile devices that allow users to engage in a wide variety of activities [55] such as sending messages, browsing the internet, and taking photos, and past research suggests these uses differ in their effects. Thus, it may be the case that the associations we observed are stronger, weaker, or do not hold for certain patterns of use. Future research of this kind should consider the ways associations differ depending on the type of smartphone use adolescents are engaged in. Third, future work should explore the possibility that the associations observed in this study might differ as a factor of stable individual differences that were not modeled in our analyses. For example, research has shown that adolescent girls have different motivations for smartphone use than adolescent boys [56], and adolescents' propensity to develop harmful or problematic smartphone use is related to socioeconomic status, parental relationships, and other factors [41]. Further, it known that media effects generally tend to vary as a factor of individual differences [57] By exploring the ways that moment-to-moment effects of smartphone use differ across different demographic groups, future studies may uncover reasons that some adolescents develop problematic patterns of smartphone use while others do not.

Fourth, the analysis used for this study was limited because it did not consider how effects of smartphone use on mood might vary over time. Time is known to be an important consideration of media effects because these effects are sometimes delayed [58] or build gradually over repeated exposures [59]. The present study was concerned with the possibility of transient, in-the-moment associations between smartphone use and mood, but we suggest that future work investigate the possibility of more complex longitudinal relationships. Specifically, we advise the use of multilevel cross-lagged panel models, which have been used previously to

effectively identify patterns in the appearance and duration of media effects [60]. Similarly, we recommend that future studies attend more to the possibility of moment-to-moment confounders. For example, it is possible that participants in our study reported higher moods while using their phones partly because phone use coincided with periods of leisure time.

Finally, this study demonstrated that real-world adolescent smartphone use is associated with the sort of consistent, reliable shifts in mood states thought to be sought by users seeking mood modification. This could imply that adolescents use smartphones to manage or modify their moods, but our protocol did not allow us to draw conclusions regarding adolescents' motivations. Our interpretation of these results aligns with prominent media psychology theories and with the results of past research, but there are still credible alternative explanations. For example, it is conceivable that the positive associations we observed between smartphone use and mood were incidental to adolescents' motivations or that adolescents actively sought mood modifications in some instances but experienced them incidentally in others. Given the importance of mood modification to theories of problematic smartphone use, we strongly encourage future research that considers adolescents' motivations for smartphone use while assessing the consequences of that use in real time.

## Conclusions

Using real-time, *in-situ* measures of adolescent smartphone use and mood, this study found that adolescents report better moods when using their phones and report mood improvements during phone use. These findings are among the first to demonstrate that smartphones have reliable, short-term associations with adolescent mood. The observation that these associations is positive implies that adolescents might use smartphones for the purpose of mood management or mood modification.

## Author Contributions

**Conceptualization:** Matt Minich, Megan Moreno.

**Formal analysis:** Matt Minich.

**Funding acquisition:** Megan Moreno.

**Methodology:** Matt Minich.

**Project administration:** Megan Moreno.

**Resources:** Megan Moreno.

**Supervision:** Megan Moreno.

**Writing – original draft:** Matt Minich.

**Writing – review & editing:** Megan Moreno.

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
