## [Editor Report · Decision Letter 0]

23 May 2023

PONE-D-23-00811Real-world adolescent smartphone use is associated with improvements in mood: An ecological momentary assessment studyPLOS ONE

Dear Dr. Minich,

Thank you for submitting your manuscript to PLOS ONE. After careful consideration, we feel that it has merit but does not fully meet PLOS ONE’s publication criteria as it currently stands. Therefore, we invite you to submit a revised version of the manuscript that addresses the points raised during the review process.

We look forward to receiving your revised manuscript.

Kind regards,

Ricardo Limongi

Academic Editor

PLOS ONE

2. We noted in your submission details that a portion of your manuscript may have been presented or published elsewhere.

[The data analyzed for this paper were also analyzed for a manuscript that is currently under review at another journal. These analyses were conducted independently, produced different results, and address different theoretical questions --- it is the belief of these authors that each makes a distinct contribution to the scientific literature. As requested, the other manuscript has been uploaded along with this submission.]

Please clarify whether this publication was peer-reviewed and formally published. If this work was previously peer-reviewed and published, in the cover letter please provide the reason that this work does not constitute dual publication and should be included in the current manuscript.

Additional Editor Comments:

The manuscript entitled " Real-world adolescent smartphone use is associated with improvements in mood: An ecological momentary assessment study "explores the relationship between adolescent smartphone use and mood using a longitudinal methodology that measured both in real time. Through a quantitative approach, the main result is that adolescent smartphone use is positively associated with mood.

To contribute to the advancement of the manuscript, considerations will be presented as follows:

1. For a better detailing into the introduction as a contribution, relevance, and theoretical cut, the opening could be organized using paragraphs as follows:

(i) Common ground (what we know in the literature): The author must present the basic premises of the literature under analysis to establish a starting point for the reader.

(ii) Complication (what are the limitations – gaps – of the literature): The author must expose missing elements or flaws in the literature (incoherent, incomplete, or contradictory theories) to explain some phenomenon that it should explain. Practical, real-world examples can help expose the limitations of the literature.

(iii) Concern: The author must show that the limitation of the literature is something relevant to be studied. More than a trivial limitation is required to motivate a search. The author should write about How and Why restriction impairs our understanding of a phenomenon.

(iv) Course of Action: The author should show the course of action that may resolve the central complication. To resolve the limitation, the author must develop a theory or refine an existing theory. To this end, the author can (i) suggest new constructs; (ii) model new relationships between constructs; (iii) explore the theoretical process; (iv) develop a typology. (v) Contribution: The author must explain how his course of action can contribute significantly to the existing theory (that which has been described on the common ground).

2. The authors could address a formal objective at the end of the introduction; in this sense, the reader will understand better the course perspective.

3. The limitations presented could be discussed in a longitudinal perspective aligned with state of the art about the central construct: the mood in a specific context, an adolescent.

4. The research questions appear on page 8 and could be addressed with the limitations and theory discussion.

5. the period the authors choose to collect data needs to be clarified.

6. Excellent perspective in choosing longitudinal and multilevel modeling procedures.

7. For better replicability of the study, the authors could present a flowchart of the analytical proposal.

8. When presenting the experimental perspective, it would be interesting to discuss the modeling of the levels of the variables and if the proposed scales were updated based on the literature through validation.

9. Could approach the analysis from the gender of the respondent for a comparison of different perspectives and effects.

---

## [Author Response · Author response to Decision Letter 0]

30 Jun 2023

Responses are in an uploaded document

---

## [Decision Letter · Decision Letter 1]

26 Dec 2023

PONE-D-23-00811R1Real-world adolescent smartphone use is associated with improvements in mood: An ecological momentary assessment studyPLOS ONE

Dear Dr. Minich,

Thank you for submitting your manuscript to PLOS ONE. After careful consideration, we feel that it has merit but does not fully meet PLOS ONE’s publication criteria as it currently stands. Therefore, we invite you to submit a revised version of the manuscript that addresses the points raised during the review process.

**ACADEMIC EDITOR:**
 Dear Authors,

I recently had the opportunity to read your new version and the feedback of reviewers. Firstly, I would like to commend the effort and dedication invested in the study, addressing an extremely relevant topic at the intersection of technology and mental health. However, I would like to offer some main points to consider in the next round: 

1. Methodological Clarity: There are moments in the document where the description of the employed methods is somewhat vague or confusing. Specifically, the modeling of random intercepts and slopes for smartphone use and duration of use could benefit from a more detailed explanation. This will help readers better understand the rationale behind the methodological choices and how they affect the interpretation of the results.

2. Detailing of Results: Including detailed tables of the model results, especially the parameters of random effects, would significantly strengthen the results section. Additionally, clarifying whether Figure 1 is based on raw scores or marginal predictions would provide a more precise understanding of the findings.

3. Discussion of Limitations and Confounding Variables: While you acknowledge some limitations, a more comprehensive discussion of possible unmeasured confounding variables that may influence the observed correlation between phone use and mood would be valuable. This would not only deepen the discussion but also provide a clearer path for future research.

4. Consider External Effects: The analysis appears not to include considerations of the effects of the day of the week or time of day. These factors could significantly impact adolescents' mood and behavior and, therefore, could be important variables to explore in future analyses.

5. Technical Specifications: The absence of specification of the software used for analysis can affect the replicability of the study. Providing these detailed pieces of information would help other researchers better understand and potentially reproduce your work.

6. Expansion of Discussion: The discussion on the use of the phone as a mood management tool and potentially addictive behaviors is a strong aspect of the paper. Expanding this discussion to include practical implications and recommendations for future interventions or policies could increase the practical impact of the study.

In summary, your study makes a significant contribution to the field and has the potential to inform future research, policies, and practices. I hope that the feedback form reivewers could be considered in the next round.

Sincerely,

We look forward to receiving your revised manuscript.

Kind regards,

Ricardo Limongi

Academic Editor

PLOS ONE

Journal Requirements:

Reviewers' comments:

Reviewer's Responses to Questions

**Comments to the Author**

1. If the authors have adequately addressed your comments raised in a previous round of review and you feel that this manuscript is now acceptable for publication, you may indicate that here to bypass the “Comments to the Author” section, enter your conflict of interest statement in the “Confidential to Editor” section, and submit your "Accept" recommendation.

Reviewer #1: (No Response)

Reviewer #2: (No Response)

2. Is the manuscript technically sound, and do the data support the conclusions?

Reviewer #1: Partly

Reviewer #2: Partly

3. Has the statistical analysis been performed appropriately and rigorously? 

Reviewer #1: I Don't Know

Reviewer #2: Yes

4. Have the authors made all data underlying the findings in their manuscript fully available?

Reviewer #1: No

Reviewer #2: No

5. Is the manuscript presented in an intelligible fashion and written in standard English?

Reviewer #1: Yes

Reviewer #2: Yes

6. Review Comments to the Author

Reviewer #1: This paper reports results from momentary assessments of smart phone use and mood among U.S. adolescents. The authors find that smart phone use was associated with increases in positive mood. The study has some strengths, however there are also limitations, some of which are acknowledged. My main concern is in the presentation of the methods and results.

Introduction

1. Although somewhat lengthy, the introduction does a good job of reviewing the literature and highlighting the limitations of previous research.

Methods

2. Measures of phone use and time on the phone were self-reported. The authors acknowledge this as a limitation. Although the immediacy of the assessment obviates any significant recall bias, other self-report bias may be present.

3. Mood before smartphone use was self-reported when the EMA captured an episode of smart phone use. This could be seen as problematic. An alternative approach could be to use mood reported at the previous EMA, although this might not capture very short-term effects.

4. Authors state: “To account for the possibility that some individuals used their phones more often than others, we modeled a random intercept for using smartphone.” and “We also modeled a random intercept and slope for length of smartphone use, for similar reasons” This is unclear, as intercepts are not attached to the independent variable. I assume the authors mean that they included a random subject intercept (which accounts for intra-person correlation of mood observations), and random slopes for smart phone use and length of smartphone use. Perhaps the authors could explain how including a random intercept accounts for variation in frequency of phone use.

5. On the analysis for RQ2, authors state “we conducted a two-tailed single-sample t-test on change in mood scores to determine whether they were significantly positive or negative.” Again this is unclear. I must assume the authors mean that they conducted a paired t-test on mood reported at EMA vs. reported mood prior to phone use. This would not adjust for intra-person correlation of repeated mood observations.

6. The analysis did not include day of week or time of day effects.

7. Software used for analysis was not specified.

Results

8. It would be helpful to include tables of the model results, including random effects parameters.

9. On figure 1, is this based on raw scores or marginal predictions?

Discussion

10. The authors do not discuss possible unmeasured confounders that may explain the correlation between phone use and mood.

11. The discussion of phone use as mood management and addictive behaviors is appropriate.

Reviewer #2: This study presents findings derived from immediate assessments of smartphone use and emotional states among American adolescents. Researchers identified a relationship between smartphone usage and an increase in positive mood. Although the study demonstrates merits, it also faces several limitations, some of which are explicitly acknowledged by the authors. My primary concern falls on the manner in which the methods and results were detailed and explained.

In the introduction, despite being somewhat lengthy, the literature review is effectively conducted, pointing out the restrictions identified in previous research. This establishes a solid background for the current study, albeit prolonging the initial section.

Regarding the methods, the authors indicate that both the phone use and the time spent were reported by the users themselves, a limitation recognized due to the possibility of self-report biases, despite the reduction of memory biases due to the immediacy of data collection. The mood before smartphone use was also reported by the users at the moment the Event-Contingent Self-Monitoring (EMA) detected device use. This method could be problematic, and an alternative suggested would be to employ the mood reported at the previous EMA to capture mood variations, although this might overlook very rapid fluctuations.

The authors mention that they modeled a random intercept to account for individual differences in the frequency of smartphone use, as well as a random intercept and slope for the duration of use, which may have been confusing due to a lack of clarity on how these intercepts relate to the independent variable. They could have offered a more detailed explanation of how this modeling helps to consider variations in frequency and duration of use among individuals.

Furthermore, the analysis for the second research question apparently involved a two-tailed single-sample t-test, though the description is ambiguous. It is presumed that the authors applied a paired t-test, comparing the mood reported at the EMA with the mood reported previously, a technique that does not account for the intra-personal correlation of repeated mood observations.

Interestingly, the analysis did not consider the effects of the day of the week or the time of day, and the authors did not specify the software used for the analysis. This might impact the interpretation of the results and the replicability of the study.

In the results, it would be beneficial to include tables detailing the model findings, especially the parameters of the random effects. There's also a question about whether Figure 1 represents raw scores or marginal predictions, a detail that could clarify the interpretation of the data.

In the discussion, the authors omitted a consideration of possible unmeasured confounding variables that could influence the observed correlation between phone use and mood. However, they aptly address the use of the phone as a mood management tool and discuss potentially addictive behaviors, a relevant aspect given the study's focus on adolescents.

In summary, while the study offers valuable insights into the relationship between smartphone use and mood among adolescents, clarity in the presentation of methods and a more comprehensive discussion on limitations and implications could further strengthen the research's contribution.

7. PLOS authors have the option to publish the peer review history of their article (what does this mean?). If published, this will include your full peer review and any attached files.

Reviewer #1: **Yes: **Mary E. Mackesy-Amiti

Reviewer #2: No

---

## [Author Response · Author response to Decision Letter 1]

23 Jan 2024

We appreciate the thoughtful comments of both reviewers, which we feel have substantially improved the manuscript. As the two reviewers have identified many similar issues, we have grouped our responses together below.

In adapting our analyses to accommodate reviewer feedback, we also excluded eight participants from our analysis due to missing data. We have thus updated the description of our data cleaning procedure (on p. 12), and descriptive statistics throughout the manuscript have been updated to reflect the removal of these participants. Removal of these participants did not meaningfully affect the results of either regression analysis. 

Item one: Risk of self-report bias

R1: “Measures of phone use and time on the phone were self-reported. The authors acknowledge this as a limitation. Although the immediacy of the assessment obviates any significant recall bias, other self-report bias may be present.”

R2: “Regarding the methods, the authors indicate that both the phone use and the time spent were reported by the users themselves, a limitation recognized due to the possibility of self-report biases, despite the reduction of memory biases due to the immediacy of data collection.”

We have expanded our description of this limitation to include the possibility of additional self-report biases: 

“Importantly, these reports may also be vulnerable to other biases associated with self-report [54]. For example, some participants may have underreported length of phone use due to social desirability pressures” (p.21)

Item two: Risk of recall bias

R1: “Mood before smartphone use was self-reported when the EMA captured an episode of smart phone use. This could be seen as problematic. An alternative approach could be to use mood reported at the previous EMA, although this might not capture very short-term effects.”

R2: “The mood before smartphone use was also reported by the users at the moment the Event-Contingent Self-Monitoring (EMA) detected device use. This method could be problematic, and an alternative suggested would be to employ the mood reported at the previous EMA to capture mood variations, although this might overlook very rapid fluctuations.”

We appreciate this insight and agree that the recalled nature of our mood before smartphone use is an important limitation of this study. As the reviewer has noted, we feel that comparison of current mood ratings with ratings given in previous assessments is equally problematic because the timing of assessments was too sparse to capture the very short-term effects this study was interested in. 

Item three: Description of random effects

R1: “Authors state: ‘To account for the possibility that some individuals used their phones more often than others, we modeled a random intercept for using smartphone.’ and ‘We also modeled a random intercept and slope for length of smartphone use, for similar reasons’ This is unclear, as intercepts are not attached to the independent variable. I assume the authors mean that they included a random subject intercept (which accounts for intra-person correlation of mood observations), and random slopes for smart phone use and length of smartphone use. Perhaps the authors could explain how including a random intercept accounts for variation in frequency of phone use.”

R2: “The authors mention that they modeled a random intercept to account for individual differences in the frequency of smartphone use, as well as a random intercept and slope for the duration of use, which may have been confusing due to a lack of clarity on how these intercepts relate to the independent variable. They could have offered a more detailed explanation of how this modeling helps to consider variations in frequency and duration of use among individuals.”

Thank you for this note, which highlights a flaw in the description of our methodology and provides an opportunity for additional clarity. We have edited the description of our model as follows:

“The model included by-participant random intercepts to account for intra-participant correlations in mood ratings, and by-participant random slopes for both using smartphone and length of smartphone use to account for participant level differences in the strength of those effects.” (p. 14)

Item four: Choice of analysis for RQ2

R1: “On the analysis for RQ2, authors state “we conducted a two-tailed single-sample t-test on change in mood scores to determine whether they were significantly positive or negative.” Again this is unclear. I must assume the authors mean that they conducted a paired t-test on mood reported at EMA vs. reported mood prior to phone use. This would not adjust for intra-person correlation of repeated mood observations.”

R2: “Furthermore, the analysis for the second research question apparently involved a two-tailed single-sample t-test, though the description is ambiguous. It is presumed that the authors applied a paired t-test, comparing the mood reported at the EMA with the mood reported previously, a technique that does not account for the intra-personal correlation of repeated mood observations.”

We appreciate this comment, which identified a way in which our analysis could be improved. In response, we have adjusted the analysis for our second research question. We now use multilevel modeling to assess the effects of phone use on changes in mood while controlling for intra-participant correlations: 

We also adjusted the structure of the data so that both current mood and mood before smartphone use were denoted as mood, with a separate time variable noting whether the mood was reported before or during smartphone use. We tested the fit of a linear mixed effects model in which mood served as the outcome variable and time of day, weekend, age, gender, race, ethnicity, free lunch, depression, time, and length of smartphone use served as predictors. This model included a by-participant random intercept to account for intra-participant correlations in mood, and by-participant random slopes for the effects of time and length of smartphone use. (p. 15) 

Item five: Inclusion of time of day and day of week effects

R1: “The analysis did not include day of week or time of day effects.”

R2: “Interestingly, the analysis did not consider the effects of the day of the week or the time of day…”

We agree that this is an important limitation of our analysis, and so we have added time of day (hour) and day of week (weekday/weekend) as fixed effects in both of our multilevel models. We have also added a section explaining these predictors: 

“Past research suggests that adolescent moods exhibit certain predictable patterns over time, with adolescents reporting mood improvement over the course of the day [41] and more positive affect on weekend days [42]. Thus, we used the timestamp associated with each assessment to denote the time of day and day of week in which the assessment was completed. Because past work notes an important distinction between weekdays and weekends, we categorized day of week into weekday (Monday – Friday) and weekend (Saturday – Sunday) days.” (p. 12)

Item six: Clarification of analysis software

R1: “Software used for analysis was not specified”

R2: “…the authors did not specify the software used for the analysis. This might impact the interpretation of the results and the replicability of the study.”

This has been clarified with the following language: 

“Analyses were conducted in R version 4.3.2 (2023-10-31) [42] using the lme4 package [43]” (p. 14)

Item seven: Presentation of model results

R1: “It would be helpful to include tables of the model results, including random effects parameters.”

R2: “In the results, it would be beneficial to include tables detailing the model findings, especially the parameters of the random effects.”

We have now added such a table for each of the two multilevel regression models. 

Item eight: Clarification about figure 1

R1: “On figure 1, is this based on raw scores or marginal predictions?”

R2: “There's also a question about whether Figure 1 represents raw scores or marginal predictions, a detail that could clarify the interpretation of the data.”

We have added language to the Figure 1 caption clarifying that this figure was created using raw scores. 

Item nine: Discussion of possible confounders

R1: “The authors do not discuss possible unmeasured confounders that may explain the correlation between phone use and mood.”

R2: “In the discussion, the authors omitted a consideration of possible unmeasured confounding variables that could influence the observed correlation between phone use and mood”

We have added the following language to our limitation section urging future attention to moment-by-moment confounders: 

“Similarly, we recommend that future studies attend more to the possibility of moment-to-moment confounders. For example, it is possible that participants in our study reported higher moods while using their phones partly because phone use coincided with periods of leisure time.” (p. 22)

---

## [Editor Report · Decision Letter 2]

25 Jan 2024

Real-world adolescent smartphone use is associated with improvements in mood: An ecological momentary assessment study

PONE-D-23-00811R2

Dear Dr. Minich,

We’re pleased to inform you that your manuscript has been judged scientifically suitable for publication and will be formally accepted for publication once it meets all outstanding technical requirements.

Kind regards,

Ricardo Limongi

Academic Editor

PLOS ONE
---

## [Editor Report · Acceptance letter]

30 Apr 2024

PONE-D-23-00811R2 

PLOS ONE

Dear Dr. Minich, 

I'm pleased to inform you that your manuscript has been deemed suitable for publication in PLOS ONE. Congratulations! Your manuscript is now being handed over to our production team.

Kind regards, 

on behalf of

Professor Ricardo Limongi 

Academic Editor

PLOS ONE